# Preventive Effects of Medium-Chain Fatty Acid Intake on Muscle Atrophy

**DOI:** 10.3390/nu17132154

**Published:** 2025-06-28

**Authors:** Madoka Sumi, Takuro Okamura, Tomoyuki Matsuyama, Tomoki Miyoshi, Hanako Nakajima, Naoko Nakanishi, Ryoichi Sasano, Masahide Hamaguchi, Michiaki Fukui

**Affiliations:** 1Department of Endocrinology and Metabolism, Graduate School of Medical Science, Kyoto Prefectural University of Medicine, Kyoto 602-8566, Japan; sumi0220@koto.kpu-m.ac.jp (M.S.); tomo10@koto.kpu-m.ac.jp (T.M.); tomokimi@koto.kpu-m.ac.jp (T.M.); tabahana@koto.kpu-m.ac.jp (H.N.); naoko-n@koto.kpu-m.ac.jp (N.N.); mhama@koto.kpu-m.ac.jp (M.H.); michiaki@koto.kpu-m.ac.jp (M.F.); 2Department of Diabetes and Endocrinology, Kyoto Okamoto Memorial Hospital, Kyoto 613-0034, Japan; 3AiSTI Science Co., Ltd., Wakayama 640-8390, Japan; sasano@aisti.co.jp

**Keywords:** coconut oil, medium-chain fatty acid, long-chain fatty acid, muscle atrophy

## Abstract

**Background/Objectives**: Medium-chain fatty acids (MCFAs), abundant in coconut oil, have attracted considerable attention in recent years owing to their potential impact on muscle atrophy. However, the mechanisms underlying their effects remain inadequately understood. This study aimed to examine the impact of coconut-oil-derived MCFAs on skeletal muscle in a mouse model administered a high-fat diet. **Methods:** C57BL/6J mice were assigned to a normal diet, lard diet, or coconut oil diet and maintained for a duration of 12 weeks. A glucose tolerance test was conducted, and biochemical parameters, muscle histological analysis, and gene expression in muscle tissue were assessed. MCFA concentrations in serum and muscle were quantified utilizing gas chromatography–mass spectrometry. An in vitro experiment was conducted by treating mouse C2C12 myotube cells with lauric acid and palmitic acid, followed by a gene expression evaluation. **Results:** Mice fed a coconut-oil-based diet exhibited reduced body weight gain and lower blood glucose and total cholesterol levels compared to those fed a lard-based diet. The coconut-oil-fed group showed increased concentrations of MCFAs in both serum and muscle tissue, along with an improvement in relative grip strength. The expression levels of proteins and genes associated with muscle atrophy were reduced in muscle tissue. These findings were corroborated in vitro using C2C12 myotube cells. **Conclusions:** Coconut oil may preserve muscle strength by increasing MCFA concentrations in serum and muscle tissue, while suppressing the expression of muscle-atrophy-related proteins and genes. These findings suggest that coconut oil may be beneficial in preventing muscle atrophy induced by long-chain fatty acids.

## 1. Introduction

In recent years, medium-chain fatty acids (MCFAs) and medium-chain triglycerides (MCTs) have attracted considerable attention for their potential preventive and therapeutic effects on various metabolic disorders, including cancer, obesity, and sarcopenia [1]. MCFAs are straight-chain saturated fatty acids with carbon chain lengths ranging from 6 to 12. Unlike conventional dietary fats that contain long-chain fatty acids (LCFAs), MCTs, which are composed of MCFAs, are transported directly to the liver via the portal vein, where they are rapidly metabolized through β-oxidation, exhibiting a lower propensity for storage as body fat [2]. Previous research has suggested that the persistent accumulation of LCFAs in muscle tissue may impair protein synthesis, induce lipotoxicity, activate immune cells, and trigger inflammation within muscle cells, ultimately contributing to muscle atrophy and the apoptosis of muscle fibers [3]. In vitro studies have demonstrated that the expression of multiple apoptosis-promoting genes is upregulated by LCFAs, with *FOXM1* (Forkhead box M1) and *PUMA* (p53-upregulated modulator of apoptosis) being key examples [4]. Additionally, pentadecanoic acid (C15), a type of LCFA, accumulates in skeletal muscle, promoting inflammation, insulin resistance, protein degradation, and muscle atrophy, ultimately reducing skeletal muscle mass [5]. In contrast, compared to long-chain triglycerides (LCTs), MCTs have been reported to contribute to improvements in muscle strength and daily functional abilities [6]. Furthermore, the combined administration of MCFAs and glucose has been suggested to be beneficial in preventing cancer-related skeletal muscle atrophy [7].

These findings indicate that MCFAs may exert beneficial effects against LCFA-induced muscle atrophy, although the underlying mechanisms have not been elucidated. This study aims to determine the effects of coconut-oil-derived MCFAs on skeletal muscle in an LCFA-induced muscle atrophy mouse model using gas chromatography–mass spectrometry (GC/MS)-based metabolomics analysis. Furthermore, the study seeks to uncover the mechanisms through which MCFAs may suppress skeletal muscle atrophy and to propose the potential utility of nutritional interventions in preventing LCFA-induced skeletal muscle atrophy.

## 2. Materials and Methods

### 2.1. Mice

All studies conducted on laboratory animals were carried out with the approval of the Animal Research Committee of Kyoto Prefectural University of Medicine, Japan (Approval No. M2024-77). Eight-week-old male C57BL/6J (wild-type) mice (*n* = 18) were obtained from Shimizu Laboratory Supplies (Shimuzu Corporation, Kyoto, Japan) and maintained under controlled, pathogen-free conditions. The mice were kept in groups of three per cage, and assigned to one of the following diets, which were administered over a 12-week period beginning at the age of 8 weeks (*n* = 6 per group): normal diet (ND)—345 kcal/100 g, 4.6% fat (CLEA, Tokyo, Japan); lard diet (D12492)—524 kcal/100 g, 20% protein, 20% carbohydrate, and 60% fat derived from lard (Research Diets, Inc., New Brunswick, NJ, USA); and coconut oil diet (D00071501)—524 kcal/100 g, 20% protein, 20% carbohydrate, and 60% fat derived from hydrogenated coconut oil (Research Diets, Inc., New Brunswick, NJ, USA). The lard-fed group was deemed to have an LCFA-rich diet, whereas the coconut-oil-fed group was considered to have an MCFA-rich diet. A detailed comparison of the high-fat content in each diet is provided in Table 1. We selected these mice as an appropriate model for studying the effects of coconut oil, as they exhibit metabolic disorders resulting from modern Westernized dietary patterns. To ensure an equal caloric intake between the lard and coconut oil diet groups, a paired-feeding protocol was employed, in which both groups were provided with identical quantities of food. The ND group was allowed to feed ad libitum. Body weight was measured weekly. At 20 weeks of age, following an overnight fast, the mice were euthanized by administering a combination of medetomidine (0.3 mg/kg), midazolam (4.0 mg/kg), and butorphanol (5.0 mg/kg) [8]. All researchers had knowledge of the experimental setup, and the study was conducted without blinding. No mice were excluded from the analysis in any of the experimental groups.

#### 2.1.1. Sample Size Estimation

Sample size estimation was performed based on a power analysis for a two-sample independent *t*-test. The primary outcome measure was the ratio of grip strength to body weight. The mean and standard deviation (SD) for the lard diet group were 0.060 and 0.002, respectively, while those for the coconut oil diet group were 0.084 and 0.010. The effect size (Cohen’s d) was calculated using the pooled standard deviation, with a significance threshold (α) of 0.05 and a statistical power (1 − β) of 0.8. The required sample size per group was determined to be 6 using the solve power function from the statsmodels package in Python (version 3.6+).

#### 2.1.2. Analytical Procedures and Glucose and Insulin Tolerance Tests

An intraperitoneal glucose tolerance test (iPGTT) was conducted in 19-week-old mice, after a 16 h fast, using a glucose dose of 2 g/kg of body weight. Blood glucose levels were measured at specified time points utilizing a glucometer, with the blood samples obtained from a drop of blood. In addition, an insulin tolerance test (ITT) was performed following a 4 h fast, using a dose of 0.5 U/kg of body weight. The areas under the curve were calculated for both the iPGTT and ITT.

#### 2.1.3. Biochemistry

Blood was drawn via cardiac puncture from fasted mice at the point of sacrifice. Total cholesterol (T-Cho) and triglyceride (TG) levels were assessed. Biochemical analyses were conducted by the FUJIFILM Wako Pure Chemical Corporation (Osaka, Japan).

#### 2.1.4. Grip Strength

Grip strength was measured using a digital force gauge (model DS2-50N, Imada Co., Ltd., Aichi, Japan) when the mice were 19 weeks old. Grip strength was assessed in three successive trials spaced 1 min apart, with normalization to body weight.

#### 2.1.5. Histology of the Plantaris and Soleus Muscle

The plantaris and soleus muscles tissues were excised, fixed in 10% buffered formalin, and embedded in paraffin. Thin muscle tissue sections were meticulously prepared and subjected to hematoxylin and eosin staining, and photographed using a BZ-X710 fluorescence microscope (Keyence Corporation, Osaka, Japan). Two random images were taken for each sample. The myofiber cross-sectional area was quantified and carried out using ImageJ software (version 1.53k; National Institutes of Health, Bethesda, MD, USA).

#### 2.1.6. Determination of MCFAs Levels in the Serum and Muscle

The metabolomic analysis of MCFA concentrations in serum and muscle was conducted using solid-phase dehydration derivatization (AiSTI Science, Wakayama, Japan) followed by GC/MS (Agilent Technologies, Santa Clara, CA, USA). Peak detection and identification (based on retention index and spectral similarity) were performed using MSDIAL version 4.9. Based on the area values obtained, the mean of the ND group was set to 1, and ratios were calculated for the other two groups.

#### 2.1.7. Protein Extraction and Western Blot Analysis

The homogenization and extraction of the gastrocnemius muscles were performed using radioimmunoprecipitation assay (RIPA) lysis buffer [ATTO, Tokyo, Japan; 50 mmol/L Tris (pH 8.0), 150 mmol/L NaCl, 1.0% NP-40, 0.5% deoxycholate, and 0.1% sodium dodecyl sulfate (SDS)] containing a protease inhibitor cocktail (BioVision, Milpitas, CA, USA). Protein levels were quantitatively assessed utilizing a NanoDrop Lite spectrophotometer (Thermo Fisher Scientific, Waltham, MA, USA) as per the manufacturer’s guidelines. Total protein was resolved using 12% sodium dodecyl sulfate–polyacrylamide gel electrophoresis (SDS-PAGE), subsequently blotted onto membranes, and analyzed by Western blot analysis following established experimental procedures. The ChemiDoc™ Touch Imaging System (Bio-Rad, Hercules, CA, USA) was utilized to capture images of protein bands. Protein samples (10–20 µg) were incubated overnight at 4 °C with monoclonal primary antibodies against *MuRF1* (1:500) or *Gapdh* (1:3000), prepared in EzBlock Chemi (ATTO, Osaka, Japan). *MuRF1*, or *Trim63*, functions as a RING-type E3 ligase involved in the induction of muscle atrophy [9]. Following this, the membranes were incubated at room temperature (22–24 °C) for 60 min with a goat anti-rabbit IgG secondary antibody conjugated to horseradish peroxidase. Although the GAPDH signal appeared relatively faint in the coconut oil group, the same amount of total protein was loaded for all samples, and normalization was performed using the GAPDH band. The relative intensity was calculated using consistent exposure times and standardized analysis conditions across all groups. ImageJ software (NIH, Bethesda, MD, USA) was used to determine protein expression levels based on optical density. Santa Cruz Biotechnology (Santa Cruz, CA, USA) was the supplier for all antibodies used in this research.

#### 2.1.8. Gene Expression Analysis in the Muscle

The plantaris and soleus muscles were collected from mice subjected to a 16 h fasting period and rapidly snap-frozen in liquid nitrogen. The tissues were then homogenized in chilled QIAzol Lysis reagent (Qiagen) using a ball mill at 4000 rpm for 2 min, and total ribonucleic acid (RNA) was isolated as per the manufacturer’s guidelines. RNA concentration and quality were assessed using a Qubit RNA Assay (Invitrogen, Thermo Fisher Scientific, MS, USA). First-strand complementary DNA (cDNA) was synthesized from 0.5 μg of extracted total RNA using the High-Capacity cDNA Reverse Transcription Kit (Applied Biosystems, Waltham, MA, USA), following the instructions provided by the manufacturer. Utilizing the TaqMan Fast Advanced Master Mix (Applied Biosystems) and standard protocols, *Trim63* mRNA levels in plantaris and soleus muscles were quantitatively analyzed via real-time reverse transcription polymerase chain reaction (RT-PCR). The thermal cycling protocol was 2 min at 50 °C, 20 s at 95 °C, followed by 40 cycles of 1 s at 95 °C and 20 s at 60 °C. Gene expression was normalized against *Gapdh* threshold cycle (Ct) values and analyzed by the comparative 2^−ΔΔCt^ method. Expression in the ND group was defined as 1.0. Three mice per group were analyzed, with each RT-PCR reaction performed in triplicate. The following TaqMan Gene Expression Assays (Applied Biosystems) were utilized: *Trim63* (*MuRF1*): Mm01185221_m1, *Gapdh* (endogenous control): Mm99999915_g1.

#### 2.1.9. Culturing of Murine Myocytes of Skeletal Muscle

C2C12 myotube cells (mouse myoblast cell line; KAC Co., Ltd., Kyoto, Japan) were plated at a density of 2 × 10^4^ cells/well in 24-well plates for RT-PCR, and 1 × 10^5^ cells/well in 100 mm dishes for Western blotting. The cells were cultured in Dulbecco’s modified Eagle’s medium supplemented with 10% fetal bovine serum at 37 °C in a humidified atmosphere with 5% CO_2_ (day 0). The culture medium also contained 1% penicillin–streptomycin. The medium was replaced every other day. On day 8, to simulate the conditions of the lard and coconut oil diets in C2C12 myotube cells, the LCFA group was supplemented with 150 μM of palmitic acid (C16) per 1 mL of medium, while the LCFA+MCFA group was supplemented with 150 μM of palmitic acid (C16) and 30 μM of lauric acid (C12) per 1 mL of medium. The control group was cultured without supplementation. Day 9 was selected for the evaluation of myotube cells in all experimental groups.

#### 2.1.10. Protein Extraction and Western Blot Analysis of C2C12 Myotube Cells

Whole C2C12 myotube cell lysates were prepared using RIPA buffer (ATTO, Tokyo, Japan; 50 mmol/L Tris (pH 8.0), 150 mmol/L NaCl, 1.0% NP-40, 0.5% deoxycholate, and 0.1% SDS) with added protease inhibitor cocktail (BioVision, Milpitas, CA, USA) to prepare total cell lysates. Using a NanoDrop Lite spectrophotometer (Thermo Fisher Scientific, Waltham, MA, USA), protein concentrations were assessed as per the manufacturer’s guidelines. The proteins were then resolved on a 12% SDS-PAGE, transferred to membranes, and analyzed by Western blotting following standard procedures. Detection of protein bands was performed using the ChemiDoc™ Touch Imaging System (Bio-Rad). Protein expression levels were quantified by measuring optical density using ImageJ software, with the optical density of *MuRF1* normalized to that of *Gapdh* to facilitate comparative analysis across samples. Protein extracts containing 10–20 μg of protein were incubated overnight at 4 °C with primary antibodies: *MuRF1* (1:500) or *Gapdh* (1:3000), diluted in EzBlock Chemi (ATTO, Osaka, Japan). The membranes were subsequently incubated for 30 min at room temperature (22–24 °C) with goat anti-rabbit IgG secondary antibodies conjugated to horseradish peroxidase (Bio-Rad), diluted in EzBlock Chemi. Santa Cruz Biotechnology (Santa Cruz, CA, USA) was the supplier for all antibodies used in this research.

#### 2.1.11. Gene Expression Analysis in C2C12 Myotube Cells

Gene expression in C2C12 myotube cells was assessed on day 9. After removing the culture medium, the cells were detached using pipettes and homogenized in ice-cold QIAzol Lysis Reagent. Total RNA was isolated as per the manufacturer’s methodology. The High-Capacity cDNA Reverse Transcription Kit (Applied Biosystems) was employed to synthesize cDNA from RNA, employing oligodT and random hexamer primers as recommended by the manufacturer. The reaction proceeded at 37 °C for 120 min and was concluded by heating at 85 °C for 5 min. The expression levels of two genes associated with muscle atrophy, *Trim63* and *Fbxo32*, were measured by RT-PCR utilizing the TaqMan Fast Advanced Master Mix (Applied Biosystems). The thermal cycling protocol consisted of an initial step at 50 °C for 2 min and 95 °C for 20 s, followed by 40 cycles of 95 °C for 1 s and 60 °C for 20 s. Gene expression levels were normalized against *Gapdh* Ct values and analyzed employing the 2^−ΔΔCt^ comparative method. Expression in ND mice was attributed a relative value of 1.0. Data were obtained from three mice per group, with each RT-PCR reaction performed in triplicate. The following TaqMan Gene Expression Assays (Applied Biosystems) were utilized: *Trim63* (*MuRF1*): Mm01185221_m1, *Gapdh* (endogenous control): Mm99999915_g1, *Fbxo32* (Atrogin-1): Mm00499523_m1.

### 2.2. Statistical Analysis

JMP software (version 14.0; SAS Institute, Cary, NC, USA) was used to perform the statistical analysis. Group differences were assessed via one-way analysis of variance (ANOVA). Significance was assumed for *p*-values under 0.05. All graphical representations were produced with GraphPad Prism (version 10.2.2; San Diego, CA, USA).

## 3. Results

### 3.1. The Coconut Oil Diet Group Exhibited Reduced Body Weight Gain and Lower Glucose Levels Compared to the Lard Diet Group

From 8 to 20 weeks of age, body weight, food intake, iPGTT, and ITT were assessed in mice fed the ND, lard, or coconut oil diet. The coconut oil diet group exhibited an increase in body weight compared to the ND group, whereas weight gain was lower than that in the lard diet group. However, there was no significant difference in the food intake between the lard and coconut oil diet groups (Figure 1A,B). Glucose tolerance was assessed using iPGTT and ITT. The coconut oil group showed higher glucose levels compared to the ND group, but the glucose levels were lower than in the lard diet group (Figure 1C–F).

### 3.2. The Coconut Oil Diet Group Exhibited a Decrease in T-Cho Levels and a Reduction in Fat Mass Compared to the Lard Diet Group

Serum lipid levels were evaluated. The T-Cho levels in the coconut oil diet group were significantly increased in the coconut oil group relative to the ND group, yet remained significantly reduced compared to the lard diet group (Figure 1G). Analysis revealed no meaningful differences in serum TG levels among the three groups (Figure 1H). Fat weight (sum of inguinal and epididymal white adipose tissues) was measured in mice from all groups at the time of sacrifice. The coconut oil diet group exhibited an increase in relative fat weight compared with the ND group; however, it was lower than that in the lard diet group (Figure 1I).

### 3.3. The Coconut Oil Diet Group Exhibited Higher Grip Strength than the Lard Diet Group

Histological evaluations were conducted to examine the impact of coconut oil on muscle tissue. The relative grip strength was greater in the coconut oil diet group compared to both the ND and the lard diet group. Additionally, the normalized grip strength in the ND group was markedly greater than that observed in the lard diet group (Figure 2A). Exemplary histopathological images of the muscle tissue are shown in Figure 2B. The scale bars represent 100 μm. The images show the general morphology of muscle fibers in each group. No significant structural abnormalities or degeneration were observed in any group. No statistically significant differences were observed in the actual weight of the plantaris muscles among the three experimental groups (Figure 2C). The relative weight of the plantaris muscles was significantly higher in the ND group compared to the lard-fed group, but no significant differences were detected in comparisons between the other groups (Figure 2D). Group comparisons indicated no meaningful divergence in the cross-sectional area of the plantaris muscle (Figure 2E). Likewise, both the actual and normalized weights of the soleus muscle showed no significant variation among the three groups (Figure 2F,G). Likewise, no appreciable differences were detected in the cross-sectional area of the soleus muscle (Figure 2H).

### 3.4. The Coconut Oil Diet Group Exhibited Higher Concentrations of MCFAs in Both Serum and Muscle Compared to the Lard Diet Group

The levels of MCFAs in both serum and muscle were analyzed using a GC/MS system, and the relative ratio of lauric acid (C12) is presented in Figure 3A,B. The levels of lauric acid (C12) in both the serum and muscle were higher in the coconut oil diet group than in the ND and the lard diet group; however, no meaningful differences were detected between the ND and the lard diet group.

### 3.5. The Coconut Oil Diet Group Exhibited Lower Expression Levels of a Protein Implicated in Muscle Atrophy and Gene in Muscle Tissue Compared to the Lard Diet Group

An analysis of protein expression levels was conducted in the plantaris muscle. Western blot analysis revealed that *MuRF1* signaling was inactivated in the coconut oil diet group compared to both the ND and the lard diet group; however, no significant differences were detected the ND or the lard diet group (Figure 4A,B). The expression of *Trim63*, a gene associated with muscle atrophy, was higher in the coconut oil group compared to the ND group, but lower than that in the lard diet group (Figure 4C).

### 3.6. The Coconut Oil Diet Group Exhibited Lower Expression Levels of Proteins and Genes Implicated in Muscle Atrophy in C2C12 Myotube Cells Compared to the Lard Diet Group

The expression of proteins in C2C12 myotube cells was examined. Western blot analysis revealed that *MuRF1* signaling was inactivated in the LCFA+MCFA group (which was supplemented with 150 μM of palmitic acid (C16) and 30 μM of lauric acid (C12) per 1 mL of medium) relative to the control group (which was not supplemented with any additional components) and the LCFA group (which was supplemented with 150 μM of palmitic acid (C16) per 1 mL of medium); however, the difference between the control and the LCFA group did not reach statistical significance (Figure 5A,B). The expression of *Trim63* and *Fbxo32* was upregulated in the LCFA+MCFA group relative to the control group, but lower than that in the LCFA group (Figure 5C,D). These findings further support the role of MCFAs in mitigating LCFA-induced muscle atrophy.

## 4. Discussion

Mice consuming a coconut-oil-based diet exhibited reduced body weight gain and lower glucose and T-Cho levels compared to those on a lard-based diet. The coconut-oil-fed mice showed elevated MCFA concentrations in both serum and muscle tissue, along with improved relative grip strength, in contrast to the lard-fed group. Muscle tissues from coconut-oil-fed mice showed decreased expression of muscle-atrophy-related genes and proteins. Similar findings were replicated in C2C12 myotube cells, which served as an in vitro model to simulate the metabolic environment of a coconut oil diet.

Growing attention has recently been directed toward the potential role of MCFAs, abundant in coconut oil, in counteracting muscle atrophy induced by the intake of a high-fat diet. A previous study suggested that the combination of lauric acid (C12) and glucose may exert a preventive effect against cancer-related skeletal muscle atrophy [7]. Lauric acid (C12) has been proposed to function as an efficient energy substrate that enhances aerobic endurance and muscle strength by regulating glucose and lipid metabolism [10].

In the present study, although no significant differences in food intake were observed between the coconut oil and lard diet groups, the coconut oil group exhibited reduced body weight and fat gain compared to the lard-fed group. Additionally, a significant reduction in glucose levels was observed in the coconut-oil-fed group. These findings are consistent with those of previous studies demonstrating that coconut oil and MCFAs contribute to body fat reduction, thereby suppressing high-fat diet (HFD)-induced weight gain and improving insulin sensitivity [11]. Moreover, Nonaka et al. reported that replacing lard with various MCTs—the triglyceride forms of MCFAs, including caprylic triglycerides (TriC8), capric triglycerides (TriC10), and lauric triglycerides (TriC12)—resulted in lower T-Cho concentrations across all MCT diet groups compared to the lard-fed group. According to a previous study, the TriC12-fed group exhibited significantly higher TG levels than the lard group [12]. Although no significant differences in TG levels were found between the lard and coconut oil diet groups in the current study, the coconut oil group showed a significantly lower concentration of T-Cho. The discrepancy regarding TG findings between previous studies and the current study remains unclear; however, variations in feeding duration and lipid composition may be contributory factors.

Guo et al. suggested that lauric acid (C12) enhances muscle strength by regulating glucose and lipid metabolism [10]. In the current study, while muscle weight and cross-sectional area exhibited no appreciable differences between the coconut oil and lard diet groups, the coconut oil group showed a marked increase in relative grip strength and higher concentrations of lauric acid (C12) in serum and muscle tissue. These findings align with those reported in earlier studies. One of the proposed mechanisms underlying muscle atrophy involves the muscle-specific E3 ubiquitin ligase *MuRF1*. *MuRF1*, initially identified as a myofibrillar protein, is believed to regulate the kinase domain of titin, a major sarcomeric protein [13]. Additionally, *MuRF1* plays a crucial role in skeletal muscle atrophy by mediating the degradation of muscle proteins through the ubiquitin–proteasome system, a key proteolytic pathway [14]. A previous study reported that, compared with LCT intake, MCT consumption suppressed *MuRF1* protein expression in the soleus muscle [15]. Supporting these observations, our study demonstrated a significant suppression of *MuRF1* and *Trim63* expression in the muscle tissue of the coconut-oil-fed group relative to the lard-fed group. These findings were also observed in a C2C12 myotube cell model, which was designed to replicate the metabolic environment of lard- and coconut-oil-based diets by utilizing palmitic acid (C16), a commonly used high-fat-induced factor, and lauric acid (C12), the most abundant fatty acid in coconut oil. In addition, we performed RT-PCR analysis to assess *Fbxo32* (Atrogin-1) gene expression in C2C12 myotube cells. The results showed that *Fbxo32* expression was significantly elevated in the LCFA group and suppressed in the LCFA+MCFA group. These findings provide further support for the muscle-atrophy-inducing effect of LCFA and the protective role of MCFA.

This study constitutes a pioneering investigation that demonstrates the effective prevention of HFD-induced muscle atrophy using MCFAs contained in coconut oil. The concentrations of MCFAs in serum and muscle tissues were directly measured using GC/MS-based metabolomics analysis. Although the current study did not thoroughly examine the ideal amount of coconut oil, establishing a suitable dosage is required prior to human application. Moreover, the analysis of muscle fiber types (slow vs. fast) was not conducted due to sample constraints, which is a limitation of this study. Immunofluorescence staining for *Trim63* was not performed in this study owing to both sample constraints and technical limitations.

## 5. Conclusions

Coconut oil could contribute significantly to the preservation of muscle strength by elevating MCFA levels in both serum and muscle, while downregulating proteins and genes associated with muscle atrophy. These results underscore the potential of coconut oil as a key dietary component for the prevention of LCFA-induced muscle atrophy and highlight its relevance in the development of novel therapeutic strategies.

## Figures and Tables

**Figure 1 nutrients-17-02154-f001:**
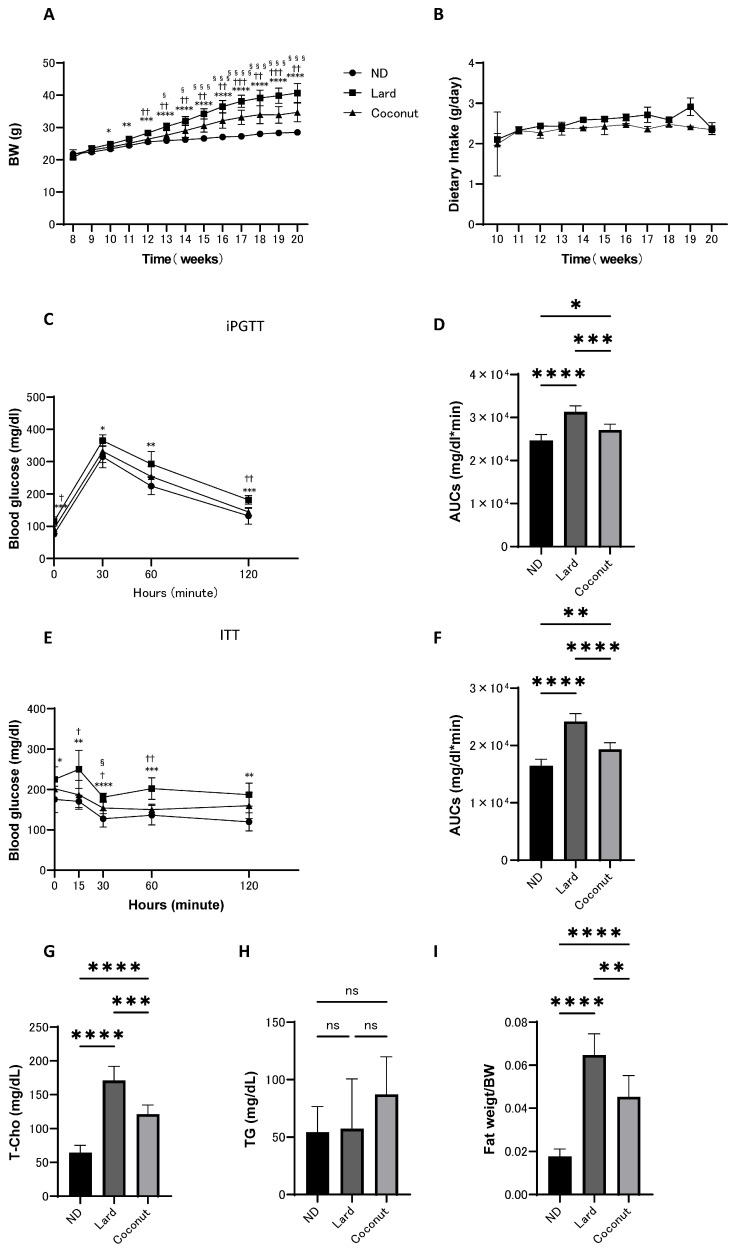
Body weight and food intake of mice maintained on ND, lard, or coconut oil from 8 to 20 weeks of age, along with iPGTT and ITT conducted at 19 weeks, and blood tests and fat weight measurements performed at 20 weeks of age. (**A**) Changes in body weight (*n* = 6). (**B**) Changes in food intake (*n* = 6). (**C**,**D**) Results of iPGTT administered at a dose of 2 g/kg body weight for mice aged 19 weeks, and AUC evaluation (*n* = 6). (**E**,**F**) Results of ITT administered at a dose of 0.5 U/kg body weight for mice aged 19 weeks, and AUC evaluation (*n* = 6). (**G**,**H**) Serum total cholesterol (T-Cho) and triglyceride (TG) levels (*n* = 6). (**I**) Relative fat weight (sum of inguinal and epididymal white adipose tissues) (*n* = 6). Data are given as the mean with its corresponding standard deviation (±SD). The comparison between ND and lard is indicated by *, between lard and coconut by †, and between ND and coconut by §. *, †, § *p* < 0.05; **, †† *p* < 0.01; ***, †††, §§§ *p* < 0.001; ****, §§§§ *p* < 0.0001; ns, not significant. AUC, area under the curve; iPGTT, intraperitoneal glucose tolerance test; ITT, insulin tolerance test.

**Figure 2 nutrients-17-02154-f002:**
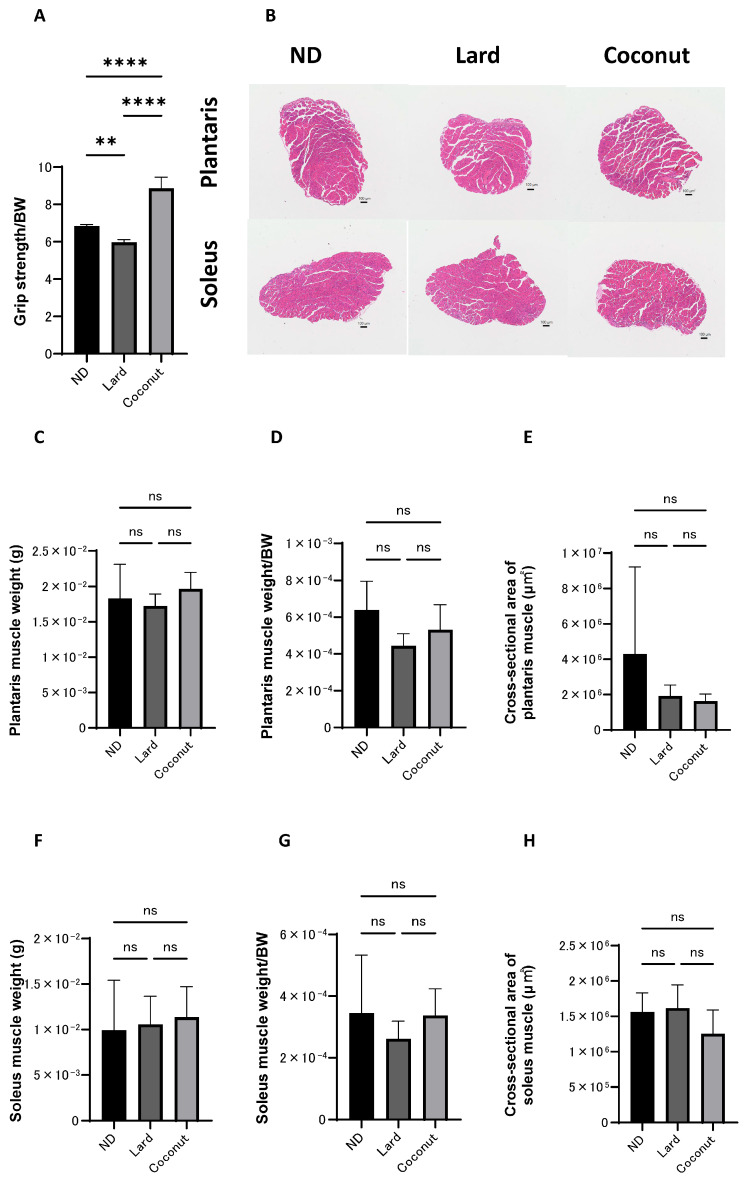
Histopathological evaluation of muscle. (**A**) Relative grip strength (*n* = 6). (**B**) Exemplary histopathological images of hematoxylin- and eosin-stained section of the plantaris and soleus muscle, harvested at 20 weeks of age. The scale bars represent 100 μm. (**C**,**D**) The actual and normalized weights of the excised plantaris muscles, assessed in mice at the age of 20 weeks (*n* = 6). (**E**) The cross-sectional area of the plantaris muscle (*n* = 6). (**F**,**G**) The actual and normalized weights of the excised soleus muscles, assessed in mice at the age of 20 weeks (*n* = 6). (**H**) The cross-sectional area of the soleus muscle (*n* = 6). Data are given as the mean ±SD, ** *p* < 0.01, and **** *p* < 0.0001; ns, not significant.

**Figure 3 nutrients-17-02154-f003:**
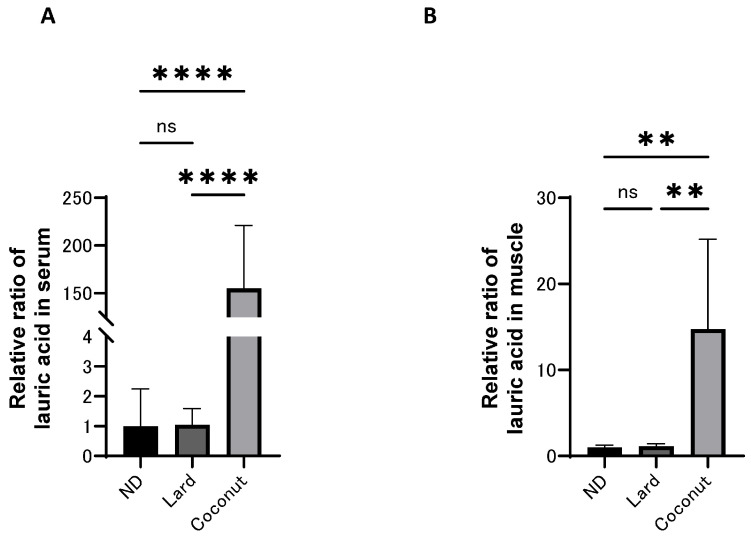
The concentration of medium-chain fatty acids in serum and muscle. (**A**) Relative ratio of lauric acid in serum (*n* = 6). (**B**) Relative ratio of lauric acid in muscle (*n* = 6). Data are given as the mean ± SD. ** *p* < 0.01 and **** *p* < 0.0001; ns, not significant.

**Figure 4 nutrients-17-02154-f004:**
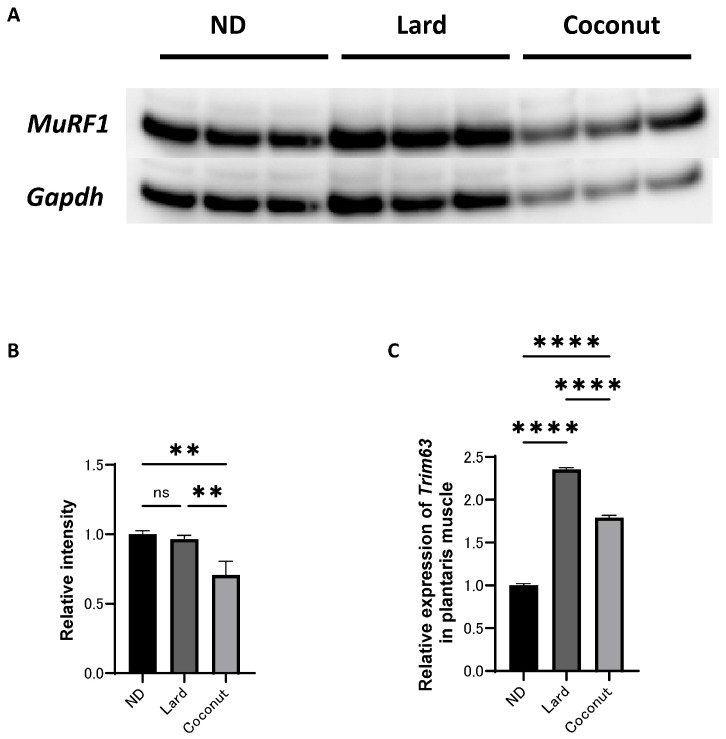
Effect of coconut on muscle atrophy proteins and gene expression in muscle. (**A**) The protein bands of *MuRF1* and *Gapdh* in the muscle analyzed by Western blot analysis. (**B**) Relative intensity of *MuRF1* normalized to *Gapdh* (*n* = 3). (**C**) Relative expression of *Trim63* in plantaris muscle (*n* = 3). Data are given as the mean ± SD. ** *p* < 0.01 and **** *p* < 0.0001; ns, not significant.

**Figure 5 nutrients-17-02154-f005:**
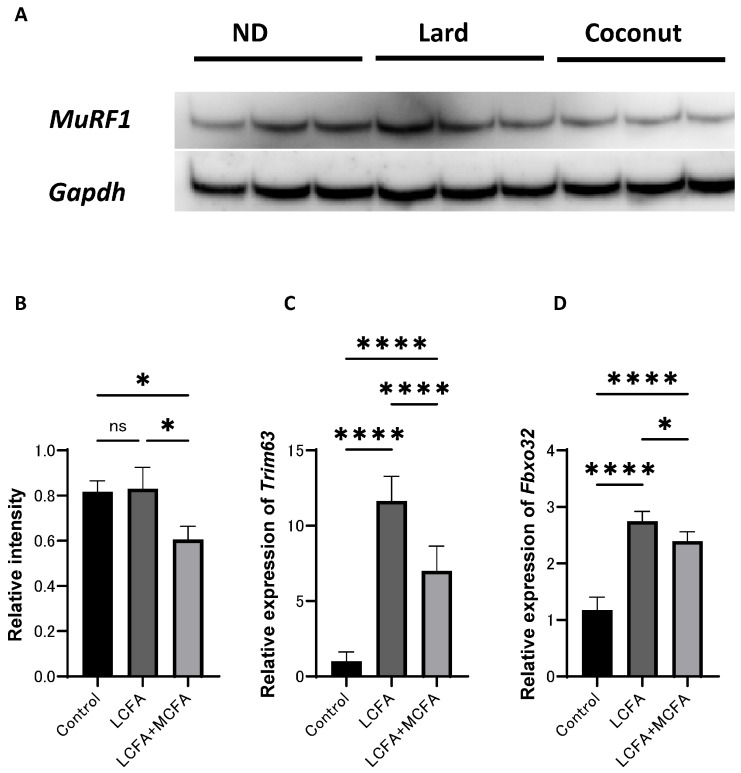
Effect of coconut on muscle atrophy proteins and gene expression in C2C12 myotube cells. (**A**) The protein bands of *MuRF1* and *Gapdh* in the C2C12 myotube cells analyzed by Western blot analysis. (**B**) Relative intensity of *MuRF1* normalized to *Gapdh* (*n* = 3). (**C**) Relative expression of *Trim63* in C2C12 myotube cells (*n* = 3). (**D**) Relative expression of *Fbxo32* in C2C12 myotube cells (*n* = 3). Data are given as the mean ±SD. * *p* < 0.05 and **** *p* < 0.0001; ns, not significant.

**Table 1 nutrients-17-02154-t001:** Detailed comparison of dietary fat content of lard and coconut oil diets.

	Lard	Coconut Oil
	(%)	(%)
Saturated	40	90
Caprylic acid (C8:0)	—	8
Capric acid (C10:0)	—	7
Lauric acid (C12:0)	—	48
Myristic acid (C14:0)	2	16
Palmitic acid (C16:0)	27	9
Stearic acid (C18:0)	11	2
Unsaturated	59	9
Oleic acid (C18:1 (n − 9))	44	7
Linoleic acid (C18:2 (n − 6))	11	2
Linolenic acid (C18:3)	—	—
Palmitoleic acid (C16:1 (n − 7))	4	—
Other	1	1

## Data Availability

All original data supporting the findings of this study are available within the article; additional information can be requested from the corresponding author.

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
