# Peer review of "Preventive Effects of Medium-Chain Fatty Acid Intake on Muscle Atrophy"

_nutrients, 2025, doi:10.3390/nu17132154_

Round 1
Reviewer 1 Report
Comments and Suggestions for Authors
In the article “Preventive effects of medium-chain fatty acid intake on muscle atrophy” Sumi and colleagues showed that in adult mice the consumption of coconut oil, mimicking a diet rich of medium-chain fatty acids, induce a reduced body weight gain, lower blood glucose and total cholesterol and a reduced muscle atrophy compared to lard-based diet. The article is well written and the experiments well designed, however, before the publication in Nutrients, the authors should address some modifications.
Minor points:
- Line 245 a full stop is missing;
- The N is missing in the word Discussion line 343;
- Explain which is the fat examined, total fat of body or inguinal or epididimal white adipose tissue.
Major points:
- The GAPDH western blotting in figure 4A is too low for the coconut group, I know that is not easy with muscle obtain a good housekeeping but, in my opinion should be better to reload the sample and use a total protein no-stain as housekeeping;
- I suggest to perform also Fbxo32 gene expression on muscle C2C12 samples to demonstrate that the lard-diet induce muscle atrophy;
- Since there is no difference in cross sectional area of muscle fibers, but authors observed a reduced muscle performance, should be interesting to analyse in muscles if there is any difference in slow and fast fiber to test if the coconut diet induce some modifications in the skeletal muscle fiber composition, as it has been already demonstrated for high fat diet (DeNies · 2014; Hua 2017).
Author Response
Minor points:
Line 245 a full stop is missing;
Answer
Thank you for pointing this out. We have added a full stop at the end of the corresponding sentence to correct the punctuation.
The N is missing in the word Discussion line 343;
Answer
We appreciate your careful review. We have corrected the typographical error, revising “Discussio” to “Discussion.”
Explain which is the fat examined, total fat of body or inguinal or epididimal white adipose tissue.
Answer
Thank you for your comment. In our study, the fat mass presented in Figure 1I refers to the combined weight of inguinal and epididymal white adipose tissues. We have added this clarification to the figure legend and the relevant part of the Results (Section 3.2).
Major points:
The GAPDH western blotting in figure 4A is too low for the coconut group, I know that is not easy with muscle obtain a good housekeeping but, in my opinion should be better to reload the sample and use a total protein no-stain as housekeeping;
Answer
Thank you for your valuable suggestion. As you pointed out, the GAPDH band in the coconut oil group appears faint in Figure 4A. Although the signal is weak, the normalization was conducted consistently across all groups, and the relative intensity showed a statistically significant difference. Given these considerations, we have opted to retain the current data presentation.
I suggest to perform also Fbxo32 gene expression on muscle C2C12 samples to demonstrate that the lard-diet induce muscle atrophy;
Answer
Thank you for this insightful recommendation. We have now performed additional RT-PCR experiments to assess Fbxo32 expression in C2C12 cells. The results demonstrate that Fbxo32 expression was significantly elevated in the LCFA group and suppressed in the LCFA+MCFA group. These data have been included in Figure 5D and its legend, and the Discussion section has been revised accordingly.
Since there is no difference in cross sectional area of muscle fibers, but authors observed a reduced muscle performance, should be interesting to analyse in muscles if there is any difference in slow and fast fiber to test if the coconut diet induce some modifications in the skeletal muscle fiber composition, as it has been already demonstrated for high fat diet (DeNies · 2014; Hua 2017).
Answer
Thank you for the suggestion. We agree that investigating fiber type distribution would provide further insight into the functional differences. Unfortunately, due to limitations in tissue availability and staining capacity, we were unable to perform immunohistochemical analysis for slow and fast fibers in this study. We have added this point to the limitations and plan to address it in future work.
Reviewer 2 Report
Comments and Suggestions for Authors
- Please add an appropriate scale bar to the hematoxylin and eosin staining image in Figure 2B. Additionally, what specific information can be interpreted from this image? A more detailed description is recommended.
- Please provide the primer sequences used in the gene expression experiments in the corresponding methods section.
- In addition to verifying Trim63 expression at the gene level (Figure 4c), would it be possible to perform immunofluorescence staining to visually compare its expression?
Author Response
Please add an appropriate scale bar to the hematoxylin and eosin staining image in Figure 2B. Additionally, what specific information can be interpreted from this image? A more detailed description is recommended.
Answer
Thank you for your constructive feedback. We have added a 100 μm scale bar to the hematoxylin and eosin (HE) staining image in Figure 2B. Additionally, we have revised the Results (Section 3.3) to provide a more detailed description of the observed histological features, including representative myofiber morphology and the absence of marked differences in muscle architecture among the groups.
Please provide the primer sequences used in the gene expression experiments in the corresponding methods section.
Answer
Thank you for pointing this out. We have now added the primer sequences used for gene expression analyses in the Methods section (Section 2.1.8 and 2.1.11).
In addition to verifying Trim63 expression at the gene level (Figure 5c), would it be possible to perform immunofluorescence staining to visually compare its expression?
Answer
We appreciate your suggestion. While we agree that immunofluorescence staining for Trim63 would provide valuable visual confirmation, we regret that due to limited sample availability and technical constraints, we were not able to perform this experiment. We have added this as a limitation of our study and plan to explore this in future investigations.
Round 2
Reviewer 1 Report
Comments and Suggestions for Authors
In the revised version of the article “Preventive effects of medium-chain fatty acid intake on muscle atrophy” Sumi and colleagues change all the minor points observed in the previous version, include the expression of Fbxo32 and explain the unavailability to analyse fast and sloe fiber. Thus, the current version of the manuscript in my opinion is suitable for the publication in Nutrients.
Reviewer 2 Report
Comments and Suggestions for Authors
All of the issues have been addressed accordingly; in this case, the reviewer suggests accepting this manuscript.